# Support or Refute: Analyzing the Stance of Evidence to Detect Out-of-Context Mis- and Disinformation

**Xin Yuan** and **Jie Guo**[*] and **Weidong Qiu** and **Zheng Huang**
School of Cyber Science and Engineering, Shanghai Jiao Tong University, China
{yuanxin, guojie, qiuwd, huang-zheng}@sjtu.edu.cn

**Shujun Li**[*]
Institute of Cyber Security for Society (iCSS) & School of Computing, University of Kent, UK
S.J.Li@kent.ac.uk

## Abstract

Mis- and disinformation online have become a major societal problem as major sources of online harms of different kinds. One common form of mis- and disinformation is out-of-context (OOC) information, where different pieces of information are falsely associated, e.g., a real image combined with a false textual caption or a misleading textual description. Although some past studies have attempted to defend against OOC mis- and disinformation through external evidence, they tend to disregard the role of different pieces of evidence with different stances. Motivated by the intuition that the stance of evidence represents a bias towards different detection results, we propose a *stance extraction network* (SEN) that can extract the stances of different pieces of multi-modal evidence in a unified framework. Moreover, we introduce a *support-refutation score* calculated based on the co-occurrence relations of named entities into the textual SEN. Extensive experiments on a public large-scale dataset demonstrated that our proposed method outperformed the state-of-the-art baselines, with the best model achieving a performance gain of 3.2% in accuracy.

## 1 Introduction

The proliferation of mis- and disinformation[1] and its erosion on democracy, justice, and public trust have increased the need for detection and intervention (Lazer et al., 2018; Islam et al., 2021). While such information consists of primarily text, the increasing popularity of non-textual data on online platforms such as images and short videos has led to more and more mis- and disinformation in different modalities beyond text (Vosoughi et al., 2018; Wang et al., 2021). In web pages such as news articles, texts and images often co-exist but they may not be correctly associated, therefore leading to the so-called out-of-context (OOC) mis- and disinformation where readers may be misled to believe some false narratives caused by such false or inaccurate text-image associations (Thomas and Kovashka, 2020). OOC mis- and disinformation have become a common phenomenon on many online platforms, and have received attention from many researchers recently (Luo et al., 2021; Aneja et al., 2023; Abdelnabi et al., 2022; Zhang et al., 2023).

A major area of research on OOC mis-/disinformation is about development of automated detection methods and construction of datasets for evaluating and comparing performances of different methods. Operations such as random matching (Jaiswal et al., 2017), manipulating named entities (Sabir et al., 2018) and strategic matching (Luo et al., 2021) have been used to automatically generate OOC mis-/disinformation samples. Rather than relying on static datasets and detection models' intrinsic ability to detect OOC mis-/disinformation (often via cross-modal inconsistencies), Müller-Budack et al. (2020) and Abdelnabi et al. (2022) also leverage available evidence on the Internet to help detect OOC mis-/disinformation. However, their methods do not adequately perform effective modeling of stance relations between evidence and the claim being checked. This can miss important information useful for detecting OOC mis-/disinformation, since a given piece of evidence supporting or refuting the claim can help detect mis-alignment between texts and images. An example can be found in Figure 1.

In this paper, to fill the aforementioned research gap on the lack of consideration of stance analysis

---

[*]Corresponding co-authors

[1]In the literature the terms "misinformation" and "disinformation" often have inconsistent definitions. In our work, we adopt the more established definitions by the United Nations (https://www.undp.org/eurasia/dis/misinformation): misinformation refers to information that is false but not created with the intention of causing harm and disinformation to information that is false and deliberately created to cause harm. Our work can be applied to both mis- and disinformation, so we will mostly use the term "mis-/disinformation".

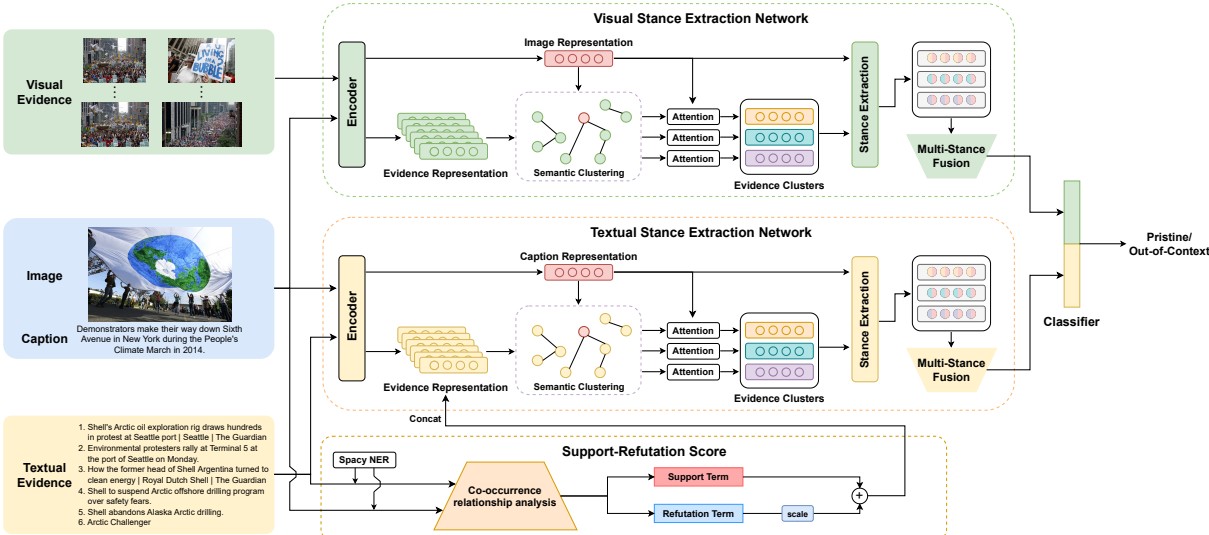

Figure 1: The architecture of our proposed method. Independent stance extraction networks are used for semantic stance comparisons between image and visual evidence, as well as caption and textual evidence, respectively. The textual extraction network incorporates support-refutation score calculated based on the co-occurrence relationship of named entities.

for OOC mis-/disinformation detection, we propose a unified framework that aims to comprehensively incorporate the stances of multiple pieces of evidence towards claims during the detection process. More specifically, for image-based claim and evidence, text-based claim (image captions in our work) and evidence, we utilize different but independent stance extraction networks (SENs) with a similar structure, which allow for cluster-specific presentations of evidence semantics and can extract and fuse multiple stances. In the textual SEN, we further emphasize the stance relationship through a support-refutation score calculated based on the co-occurrence relationship of named entities to replace the binary named entity indicator (NEI) commonly used in the literature (Tan et al., 2020; Abdelnabi et al., 2022). Our analyses based on a public large-scale dataset showed that the score follows a significantly different distribution for pristine information and OOC mis-/disinformation. Further experiments demonstrated that our SENs are more efficient at mining useful information in evidence that can be used to verify claims. The major contributions of this paper are summarized below.

- We propose a unified framework for comprehensively modeling the stance of external evidence to detect OOC mis-/disinformation.
- To the best of our knowledge, we are the first reporting the significance of stance relation for detection of OOC mis-/disinformation.

- We conducted extensive experiments to prove that our proposed method can significantly outperform state-of-the-art (SOTA) methods while being highly explainable.

The rest of the paper is organized as follows. In the next section, we overview some related work. Section 3 explains our methodology, and experimental setup and results are covered in Section 4. Section 5 concludes the paper, with the two sections after it discuss limitations and ethical considerations.

## 2 Related Work

**Fake News Detection.** Most fake news detection algorithms work solely on plain text. Some studies employ fact-checking based on knowledge bases or knowledge graph to identify fake news (Thorne et al., 2018; Zhou et al., 2019; Zhong et al., 2020). In recent years, more approaches have attempted to use multimodal feature to address the challenge of multimodal deepfakes, which requires bridging the semantic gap between multiple modalities (Imran et al., 2020). Xue et al. (2021) mapped text features and visual semantic features to the same semantic space to obtain cross-modal feature representations, and considers the consistency between them. Sun et al. (2023) designed a dual-inconsistency network to simultaneously detect cross-modal inconsistency and content-knowledge inconsistency.

**Image Repurposing and Out-of-Context Mis-**

**information.** Early studies on image repurposing check the integrity of package through reference datasets containing closely related metadata, such as MAIM (Jaiswal et al., 2017) and MEIR (Sabir et al., 2018). AIRD (Jaiswal et al., 2019) adopted an adversarial approach to simultaneously train a bad actor who forges metadata for image repurposing and a watchdog for consistency verification of images and accompanying metadata. However, it is unrealistic to assume existing an available and reliable reference datasets. Therefore, Müller-Budack et al. (2020) turned to the Internet for visual evidence to verify the images in news.

To eliminate the linguistic bias caused by manipulating named entities in the text to obtain inconsistencies, given a caption, Luo et al. (2021) retrieved an irrelevant but convincing image through CLIP (Radford et al., 2021) and other models. Abdelnabi et al. (2022) proposed the concept of multimodal cycle-consistency check that starting from image/caption, they searched for textual/visual evidence on the Internet to verify the caption/image. However, most of them only focus on evidence with similar semantics, ignoring evidence with different stance that may be essential for out-of-context misinformation detection.

**Stance Detection.** Stance detection is a classification task where the classification results are *in Favor*, *Against*, *Neither*, or other similar forms (AL-Dayel and Magdy, 2021; Hardalov et al., 2021). Zubiaga et al. (2018) demonstrated that stance is helpful for rumor detection. Some works have attempted to introduce stance detection into fake news detection, but mainly focus on the plain text situation (Thorne et al., 2018, 2019). In multimodal situations, Yao et al. (2022) generated a stance representation of evidence during the declaration verification phase to predict a truthfulness label. However, there is no work that can comprehensively introduce stance relation into the detection of out-of-context mis-/disinformation.

## 3 Methodology

### 3.1 Problem Statement

Given an image-caption pair $X = (I^c, C^c)$, which consists of an image claim $I^c$ and a caption claim $C^c$, and two sets of external evidence, namely textual evidence set $T^e = [T_1^e, \cdots, T_M^e]$ and visual evidence set $V^e = [V_1^e, \cdots, V_N^e]$, our task is to accurately predict a binary OOC label $L^c$ that indicates if $X$'s two claims are falsely associated. In other words, the task can be described as a function as follow: $(I^c, C^c, T^e, V^e) \rightarrow L^c$.

The architecture of our proposed method is shown in Figure 1. Comparisons between the different subsets of input, i.e., between the image claim and visual evidence, and between the caption claim and textual evidence, are performed by independent SENs. Our proposed method allows cluster-specific presentations of the semantics of evidence, preventing evidence with different semantic stances from being ignored. According to Tan et al. (2020), contextual information about named entities is crucial in fake news detection and they leveraged this feature using the binary NEI, which is simplistic. So our proposed method calculates a support-refutation score based on the co-occurrence relationships of named entities to emphasize the stance of each piece of evidence in the textual SEN.

### 3.2 Support-Refutation Score

The support-refutation score (SRS) is based on a simple observation: if there exist same named entities between the textual evidence and the caption claim, then they are usually related to the same or a similar context. Conversely, if the named entities that occur frequently in the textual evidence do not appear in the caption claim, the two contexts are likely to be significantly different. That is, the co-occurrence relationship of named entities can reflect the stance of textual evidence towards the caption claim. This is more evident in OOC disinformation scenarios, where the disinformation creators often tamper with named entities therefore leading to such contextual inconsistencies (Sabir et al., 2018; Müller-Budack et al., 2020).

To calculate SRSs, We use the NER module in the Spacy library (Honnibal and Montani, 2017) to perform named entity recognition on $C^c$ and $T^e$, and obtain the named entity set $E_c$ for $C^c$ and $E_{e_i}$ for the $i$-th textual evidence $T_i^e$, respectively. We obtain statistics on all $E_{e_i}$, and rank the named entity set $E_e$ by their frequencies following a descending order. Then, we calculate the SRS for the $i$-th evidence according to the following equations:

$$E_{i\&c} = E_{e_i} \cap E_c, \tag{1}$$

$$E_{i-c} = E_{e_i} - E_c, \tag{2}$$

$$\text{SRS}_i = \#E_{i\&c} - \frac{\sum_{j \in E_{i-c}, o_j \geq \tau} g(o_j)}{\zeta}, \tag{3}$$

where $o_j$ refers to the order of named entity $j$ in $E_e$, and $\tau$ is a threshold. Only named entities with an occurrence frequency in $E_e$ no less than $\tau$ will be considered, which can reduce the impact of irrelevant evidence since the quality of evidence retrieved from the Internet is not always guaranteed. Additionally, we use a scaling factor $\zeta$ that scales down as $\#E_{i\&c}$ decreases to emphasize a conflicting (mis-aligned) relation, otherwise scales up to emphasize an aligned relation. It should be noted that, due to the imperfect accuracy of the Spacy NER module, we utilize fuzzy matching based on the edit distance (Levenshtein Distance more precisely) to filter out inconsistencies such as "Barack Obama" and "Obama's" when comparing named entities.

Meanwhile, a conflicting named entity with a higher frequency should express a stronger refutation stance. So we use $g(x) = \frac{1}{a + e^{b/x}}$, a variant of the family of Logistic functions to control the impact of conflicting named entities on the SRS. $\zeta$, $a$ and $b$ are all undetermined parameters.

Although the SRS is a concise and intuitive approach suitable for many scenarios, it may not be sufficient for addressing a more complex OOC scenario where the same named entities exist but with inconsistent contexts. To address this limitation, we propose an SEN which requires full consideration of semantics.

### 3.3 Stance Extraction Network

A stance extraction network (SEN) aims to comprehensively consider the semantic stance of all evidence while preventing the neglect of important evidence by enabling the cluster-specific presentations of different semantics. Our method uses independent SENs with a similar structure for visual evidence and textual evidence.

We map claims and evidence into semantic representations in visual and textual semantic spaces. We use the same encoders as the consistency-checking network (CCN) (Abdelnabi et al., 2022). Specifically, for image claim $I^c$ and visual evidence $V^e$, we use ResNet152 (He et al., 2016) pretrained on ImageNet and ResNet50 pretrained on Places365 (Zhou et al., 2017) as two independent encoders. For caption claim $C^c$ and textual evidence $T^e$, We use Sentence-Bert (Reimers and Gurevych, 2019). We denote the representation of $I^c/C^c$ as $\hat{H}^c_{i/t}$ and the representation of $V^e/T^e$ as $\hat{H}^e_{i/t}$. It should be noted that for the encoder output of textual evidence $\tilde{H}^e_t$, we concatenate it with the SRS calculated as explained in Section 3.2 to obtain its final semantic representation:

$$\hat{H}^e_t = \text{Concat}(\tilde{H}^e_t, \text{SRS}) \qquad (4)$$

For the same of convenience and simplicity, in the following we hide the subscripts $i$ and $t$.

The semantic comparison of multiple pieces of evidence versus a single claim is a complex issue. In order to avoid the evidence with different semantics towards the claim being ignored in the attention mechanism due to its low weight, our method performs a hierarchical clustering process using the cosine distance on the claim and evidence based on the representations extracted above, and obtain the following three clusters:

1) Supporting cluster (SuC), consisting of evidence in the same cluster as the claim at a clustering threshold $\tau_s$, typically expressing support stance;
2) Representative cluster (ReC), the maximum evidence cluster at a clustering threshold $\tau_r$, which best represents the semantics of all evidence;
3) Complementary cluster (CoC), not actually a cluster, consisting of the residual evidence, often containing noise, which we retain to prevent the loss of potentially useful information.

It should be noted that all the above three clusters may be empty, and SuC and ReC may also overlap. When there are two clusters containing the maximum amount of evidence, we regard the cluster whose semantic is closer to the claim as ReC.

Referring to the way how the memory network (Sukhbaatar et al., 2015; Abdelnabi et al., 2022) processes multiple sentences, our method uses a fully connected layer to process the claim representation $\hat{H}^c$ and obtain $H^c$ as query in the attention mechanism, and the other two independent fully connected layers to process the evidence representation $\hat{H}^e$ and obtain $H^e_k$ and $H^e_v$ as key and value, as described in the equations below:

$$H^c = \sigma(W^c \hat{H}^c + b^c), \qquad (5)$$

$$H^e_{k/v} = \sigma(W^e_{k/v} \hat{H}^e + b^c_{k/v}), \qquad (6)$$

where $\sigma$ represents ReLU.

As shown in Figure 2, we complete the attention mechanism shown in Eqs. (7)–(8) for each cluster. We first compute an attention distribution

between the claim and evidence using softmax, and then treat the weighted sum as a cluster-specific semantic presentations of the evidence in this cluster. Then, we added $H^c$ to the cluster-specific semantic presentations to obtain the semantic stance representation of each cluster as follows:

$$\alpha_* = \text{Softmax}(H^c \cdot H^e_{k*}), \qquad (7)$$

$$H_* = \text{BN}(\alpha_* \cdot H^e_{v*} + H^c), \qquad (8)$$

where $* \in \{\text{SuC}, \text{ReC}, \text{CoC}\}$ represents different clusters, and BN represents batch normalization.

We fuse the semantic stance representations of the above three clusters to obtain a stance fusion representation of the overall evidence. We tried several commonly used fusion strategies including element-wise multiplication, max-pooling, avg-pooling, but the simple concatenate showed the best performance, which is described as follows:

$$H = \text{Concat}(H_{\text{SuC}}, H_{\text{ReC}}, H_{\text{CoC}}), \qquad (9)$$

$$H_{\text{fusion}} = \sigma((WH + b) + H^c). \qquad (10)$$

We added $H^c$ in Equation (10) again to emphasize the stance relation. The dimension of $W$ and that of $b$ depend on the dimension of $H^c$. Generally speaking, if $H^c \in \mathbb{R}^d$, then $W \in \mathbb{R}^{d \times 3d}$ and $b \in \mathbb{R}^d$. We set $d$ to 1024 and 768 for the visual and textual SENs, respectively.

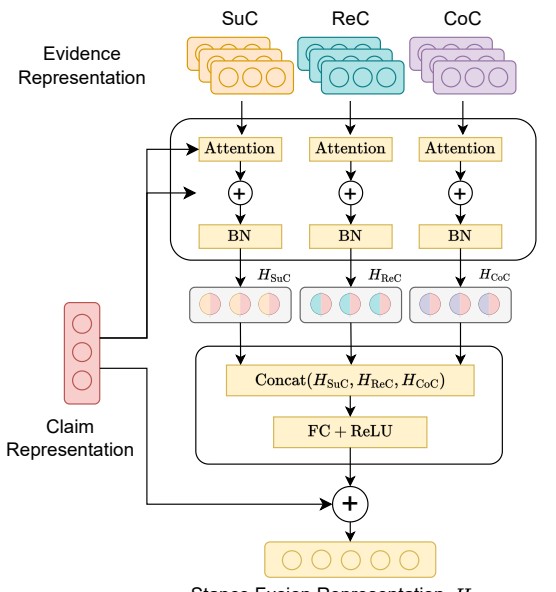

Figure 2: Stance extraction and fusion in the SEN.

Finally, we concatenate all obtained stance fusion representations from different SENs and feed them into a classifier consisting of two fully-connected layers, following the approach of Abdelnabi et al. (2022). We use the cross entropy loss to train the model.

# 4 Experiments

In this section, we explain the datasets we used, the experimental setup, and the experimental results regarding the performance of our proposed method. We also conduct qualitative analysis of the results to provide an intuitive illustration of our method.

## 4.1 Datasets

NewsCLIPpings (Luo et al., 2021) is a large-scale and currently the most challenging OOC mis-/disinformation dataset, which contains both pristine and falsified (i.e., OOC) image-caption pairs. According to certain strategies, it matches real images and real captions to simulate four OOC scenarios: (1) Both Scenarios a and b, i.e., Semantics/CLIP Text-Image and Semantics/CLIP Text-Text, are about attempts of portraying the subject of images as other named entities and use CLIP embeddings for mismatching. The main difference is that Scenario a tries to form OOC pairs by finding non-matching image-caption pairs with the highest similarity directly, while Scenario b by first finding non-matching caption-caption pairs with the highest similarity and then using the corresponding image of the second caption. (2) Scenario c, i.e., Person/SBERT-WK Text-Text, portrays the same person in a false context, such as the wrong events or places. It uses the lowest SBERT-WK text-text similarity to ensure the person appearing in different contexts. (3) Scenario d, i.e., Scene/ResNet Place, describes the event in the image as another event of the same event type.

We evaluated our proposed method on its Merged/Balanced subset, which mixes equal samples from the above four scenarios to achieve a more realistic scenario and consists of 71,072 samples for training, 7,024 for validation and 7,264 for testing.

For visual and textual evidence used to validate image-caption pairs, we used the results retrieved by Abdelnabi et al. (2022). Given a textual caption, it retrieves up to 10 pieces of visual evidence. Given an image, it retrieves textual evidence of both entity and sentence types.

## 4.2 Experimental Setup

We computed SRSs for both entities and sentence evidence, and we set the threshold $\tau = \min(2, \#E_e)$. We used binarized $\zeta$ and set $g(x) = e^{-\sqrt{x}}$, as explained in Appendix A. We used two SENs for the two different representations of image claim and visual evidence, and one SEN for sentence evidence. Due to the fact that almost all entities are general concepts or named entities composed of one or two words, they lack the complete context to express stance through semantics, resulting the small difference (about 0.1%) between using an SEN and a memory network. So we decided to use the memory network, since it has fewer parameters. Since the retrieved evidence is usually in the same field as the claim, for example, both singers' performances or politicians' speeches, we set the threshold of SuC and ReC to be the same. Specifically, we set the text clustering thresholds $\tau_s^t$ and $\tau_r^t$ to 0.500 and the image clustering thresholds $\tau_s^i$ and $\tau_r^i$ to 0.166.

To measure the performance gain from introducing stance analysis, we used all the constraints and gains in the baseline CCN (Abdelnabi et al., 2022), such as evidence filtering constraint that will reduce the accuracy rate but are closer to the real situation, and the fine-tuning CLIP model, image label and domain gain.

We conducted experiments on a computer equipped with one NVIDIA Geforce RTX 3090 GPU. The model was trained with a batch size of 64 for 60 epochs. The optimizer used is Adam (Kingma and Ba, 2014). The learning rate is a cyclical learning rate with the maximum value of 6e-5 and minimum value of 9e-6.

## 4.3 Results

We compared our method against the following SOTA baseline methods. In Table 1 we indicate whether these SOTA methods use external evidence or not.

- **CLIP** (Radford et al., 2021) generates semantically similar representations for images and text describing the same concept or event;
- **SSDL** (Mu et al., 2023) assesses intra-modal and inter-modal self-consistencies by two-phased learning strategies using self-supervised semi-supervised method;
- **DT** (Papadopoulos et al., 2023) uses transformer-based detector to process CLIP

encoded images and captions. We choose the model using ViT-B/32 like CLIP and CCN.
- **MNSL** (Zhang et al., 2023) parses text into abstract-meaning-representation graph and process it together with visual input.
- **CCN** (Abdelnabi et al., 2022) uses Internet evidence retrieved across modalities to fact-check image-caption pairs. For comparison, we choose the model using Sentence-BERT for text encoding like ours.

| | Evidence | All | Pristine | Falsified |
|---|---|---|---|---|
| CLIP | ✗ | 60.2% | 70.1% | 50.4% |
| SSDL | ✗ | 65.5% | 68.7% | 62.2% |
| DT | ✗ | 65.7% | 73.7% | 57.6% |
| MNSL | ✗ | 68.2% | 65.4% | 70.5% |
| CCN | ✓ | 83.9% | 82.8% | 84.9% |
| **Ours** | ✓ | **87.1%** | **85.5%** | **88.6%** |

Table 1: Performance on Merged/Balanced subset of our method in comparison with baselines using *Accuracy* metric.

Table 1 shows the performance of all compared models on all pairs, pristine pairs and falsified pairs. Compared with all baselines, our method achieves the best performance, improving the accuracy by 3.2% compared to the second best method. The prediction of both pristine pairs and falsified pairs has been significantly improved, with an accuracy gain of 3.7% on falsified pairs, slightly higher than the 2.7% on pristine pairs, which confirms that considering the stance of evidence will benefit detection of OOC mis-/disinformation.

Table 1 also demonstrates the importance of external evidence for detection of OOC image-caption pairs. Models using external evidence achieve significantly higher accuracy than those not. Furthermore, due to the highly realistic and plausible nature of falsified samples, models not relying on evidence have over-predicted the *Pristine* or *Falsified* label, which means it is difficult for them to mine more useful features relying solely on pairs.

We further compared the accuracy of our method with CCN on the data from the four scenarios in the Merged/Balanced subset to analyze their ability to deal with various types of OOC. As shown in Figure 3, compared with CCN, our method exhibits significant improvements for Scenarios a and d by 3.8% and 4.7%, respectively. For Scenario b, due to the use of indirect mismatching method

which is easier to detect, both CCN and our method have achieved an accuracy rate of more than 90% and the performance improvement at this time begins to be affected by the quality of the both pairs and evidence, but our model still achieved a 1.3% improvement. As discussed in Luo et al. (2021), Scenario c is the most challenging one, and our method achieves 3.0% performance improvement, indicating that our method has a better understanding ability and can comprehensively mine information beyond named entities.

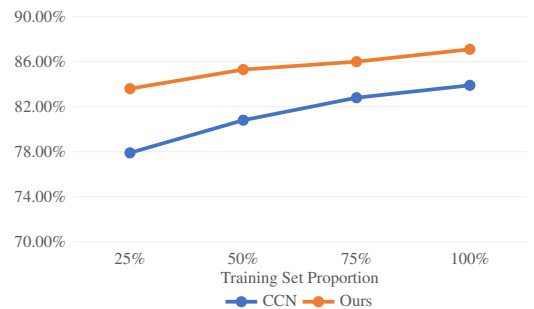

Figure 4: Performance analysis in limited data environment.

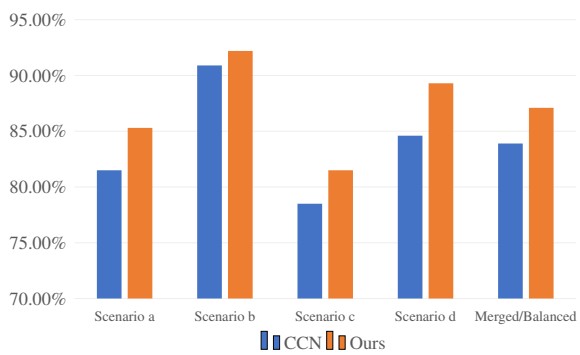

Figure 3: Performance comparison between CCN and our method in different out-of-context scenarios.

## 4.4 Performance Analysis in Limited Data Environment

To evaluate the performance stability of our method in a limited data environment, we randomly sampled the training set at varying proportions. The results, depicted in Figure 4, indicate that with only 25% of the entire training set required, our method has approached the accuracy performance of the baseline CCN on the entire training set, reaching 83.6%. Furthermore, our proposed method can have an even higher performance gain with a smaller number of training samples. Detailed data can be found in Appendix B. This finding highlights our method's superior capacity to effectively explore and utilize useful features for the OOC mis-/disinformation detection task.

## 4.5 Ablation Analysis

We investigated the impact of different components of our proposed method on the performance by defining the following variants:

**w/o SRS:** Remove the SRS for textual evidence.

**binary NEI**: Use binary NEI to replace SRS.

**w/o Vi/Te-SEN:** Use Memory Network to replace the Visual/Textual Stance Extraction Networks.

**w/o SENs:** Use Memory Network to replace all Stance Extraction Networks.

**w/o Cluster:** Remove evidence belonging to the specific cluster.

The ablation results in Table 2 confirm that both SRS and SEN are indispensable for the best performance. Compared with *w/o SRS*, the widely used *binary NEI* (Tan et al., 2020; Abdelnabi et al., 2022), although improving the detection accuracy of pristine pairs, weakens the ability to detect falsified pairs, while SRS is able to improve both. Although the SEN with both modalities may slightly impair the detection of pristine pairs (up to 1.4%), it substantially improves the detection of falsified pairs (up to 5.7%). This is consistent with our hypothesis that both SRS and SEN are intended to highlight the refuting relation while preserving the supporting relation implied in the baseline.

In order to verify the effectiveness of semantic clustering, we removed different clusters for importance analysis. As shown in Table 2, when different clusters are removed, performance decreases to varying degrees, with CoC having the least impact, as there is more noise rather than useful information. Notably, given the possibility of evidence intersection between SuC and ReC, we removed both of them at the same time, resulting in a significant performance drop.

## 4.6 Explainability Analysis

Figure 6 shows the distribution of the mean SRS for both pristine and falsified pairs, showing significant distribution differences. Our model has learned the distribution differences to better predict labels, improving the accuracy by 2.5% and 1.4% compared to *w/o SRS* and *binary NEI* respectively, as shown in Table 2. It should be noted that in Scenario c, i.e., the Person/SBERT-WK Text-Text scenario, some distributions appear bigger than 0 due to the high

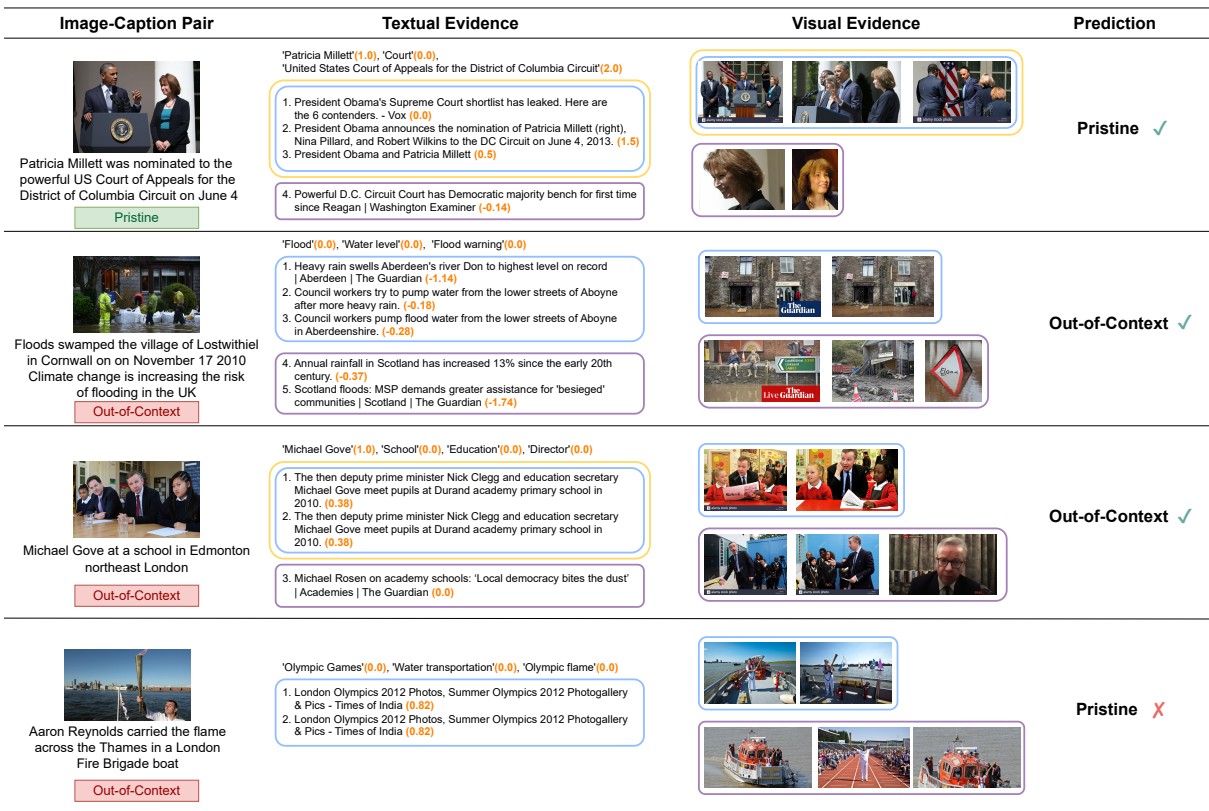

Figure 5: Some prediction examples. Yellow box represents SuC, blue box represents ReC, purple box represents CoC, and (orange) represents SRS for textual evidence. The ground truth labels are listed in *Image-Caption Pair*. We report the prediction of our method in *Prediction* and use ✓/✗ to indicate whether the prediction is correct. Only part of the evidence is shown to highlight key points.

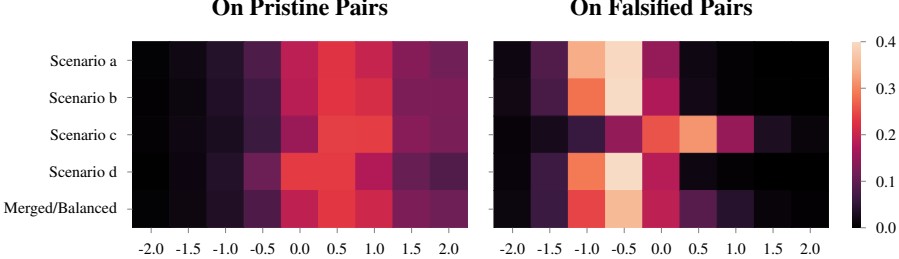

Figure 6: SRS distribution heatmap on pristine data and falsified data.

probability containing mentions of the same person. To statistically verify the observation results, two equal-sized collections of both pristine and falsified image-caption pairs were sampled, with 500 samples per type. Then we conducted a two-sample one-tail z-test on the 1,000 samples. Let $\mu_p$ be the mean SRS of pristine samples and $\mu_f$ be that of falsified samples. $H_0$ represents the null hypothesis and the alternative hypothesis is $H_1$:

$$H_0 : \mu_p - \mu_f \leq \gamma \quad (11)$$

$$H_1 : \mu_p - \mu_f > \gamma \quad (12)$$

When $\gamma$ gradually increases to 0.7, $z = 3.627$,

$p < 0.001$, the null hypothesis $H_0$ can be rejected even under a significance level ($\alpha$) as low as 0.001, indicating the observation results are statistically supported.

Figure 5 shows some prediction examples of our model. For pristine examples, such as the first one, in addition to having typically positive SRS due to the same named entities, SuC and ReC simultaneously express support semantics and often overlap. For falsified examples such as the second one, there is usually no evidence in SuC. In addition, although the textual evidence and the visual evidence have an event type similar to the image-caption pair, it

|          | **All**   | **Pristine** | **Falsified** |
|----------|-----------|--------------|---------------|
| Ours     | **87.1%** | 85.5%        | **88.6%**     |
| w/o SRS     | 84.6% | 82.1%     | 87.1% |
| binary NEI  | 85.7% | 86.9%     | 84.4% |
| w/o Vi-SEN  | 85.5% | 85.9%     | 85.1% |
| w/o Te-SEN  | 86.1% | 85.0%     | 87.3% |
| w/o SENs    | 84.9% | **86.9%** | 82.9% |
| w/o SuC     | 83.4% | 83.5%     | 83.3% |
| w/o ReC     | 85.1% | 83.9%     | 86.3% |
| w/o CoC     | 86.5% | 85.6%     | 87.4% |
| w/o SuC+ReC | 74.0% | 75.4%     | 72.6% |

Table 2: Performance comparison results of different variants.

is predicted correctly. The third one shows a situation that is difficult to judge, due to the same event context, i.e., the same person in same event type, partial textual evidence presents supporting semantics, but a low SRS and a lack of supporting visual evidence prevent the model from making wrong judgments. The last one shows a wrong prediction example, the textual evidence is "generically matched" and semantic conflicts are difficult to appear. In the visual evidence, the model may pay more attention to the characteristics of *people holding torch*, *boats*, *seas*, etc., but does not note the difference in the person due to the size.

## 5 Conclusion

We propose a unified framework capable of incorporating stance relation of evidence for automatic out-of-context mis-/disinformation detection. The support-refutation score calculates a stance related score for each textual evidence based on the co-occurrence relationship of named entities, while stance extraction networks are able to uniformly consider the stance of all evidences. Our proposed method outperforms the existing baselines and provides a new benchmark for evidence-assisted veracity assessment of image-caption pair task. Furthermore, our method offers good interpretability, allowing for a clear understanding of the role of each evidence and how it contributes to the overall decision-making process.

## 6 Limitations

We are not aware of a model for stance relation analysis based on named entities, so we propose a concise equation to model this based on the practical background. However, a unified standard or a more accurate learning model that considers semantics would be ideal for this task. Additionally, we measure named entities based on fuzzy matching, which may not account for more complex situations such as name abbreviations or relational references, etc. Although these situations are uncommon in our task scenarios, that is, short captions, it would be beneficial to consider them for more comprehensive and accurate analysis.

Multi-modal fact-checking of OOC mis-/disinformation, especially news, is a complex task due to: (1) the naturally loose and abstract correspondence between the image and the caption; (2) the multi-faceted nature of verification, including verification within modalities and between modalities; and (3) the quality of evidence is not always guaranteed. There is no single pipeline that can handle all these challenges in a unified manner so far, especially the first one, which remains a research challenge. In our future work, we plan to devise a more comprehensive model to address some of these challenges and to generalize our method to other OOC tasks, such as the more complicated one with one image and two captions addressed in Aneja et al. (2023).

## 7 Ethics Statements

In terms of ethical considerations, our method aims to address the prominent social problem of OOC mis-/disinformation by identifying characteristics that distinguish it from truthful information. We acknowledge that current detection algorithms, including our method, have failed to achieve a detection accuracy that can be completely independent of human intervention, not to mention that there are far more OOC scenarios than the four categories described in (Luo et al., 2021) and considered in our work. However, manual reviews are not feasible for the vast amount of mis-/disinformation online, and our method can serve as a fast and reasonably accurate first line of defense. For information with significant influence, such as that released by important institutions, celebrities, and politicians, our method is suitable as an auxiliary tool to aid human judgment. However, we also recognize that our method is not perfect and may have limitations such as potential biases, which should be taken into account when interpreting its results. We strive to be transparent about our methodology

and encourage ongoing ethical discussions in the field of mis/-disinformation detection.

## Acknowledgments

This work was partly supported by the National Natural Science Foundation of China under the Reference Number 61972249.

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

## A  SRS Parameter Optimization

Since we do not have any gold standards or ground truth available, we conducted a small-scale experiment to select an excellent curve for modeling support-refutation relations in SRS and illustrate the effectiveness of each item.

It should be noted that for each hyperparameter, we mainly conducted experiments based on grid search within a small and reasonable range. Considering that each parameter can theoretically take many values in a large range, this experiment is obviously not exhaustive. But our aim is to make SRS achieve representative performance and demonstrate the effectiveness of each item in SRS, rather than the optimal performance with minimal differences. Larger ranges and more detailed values

will exponentially increase time and computational resources, which is not in line with our original intention. To this end, we report the validation accuracy when selecting $\zeta$ and $g(x)$.

## A.1 Selection of Scale Factor $\zeta$

$\zeta$ controls the emphasis on corresponding or conflicting relationships in the Support-Refutation Score (SRS) calculation. We consider two approaches to setting the value of $\zeta$. The first approach, shown in Equation (13), which we call the *Proportion Setting*, is intuitive and commonly used. However, when the number of named entities in the evidence and caption overlap, i.e., $\#E_{i\&c}$ is too large, the conflicting relationship may be weakened. Therefore, we consider a second approach, *Binarization Setting*, where $\zeta$ is binarized, as shown in Equation (14).

$$\zeta = \alpha(\#E_{i\&c} + 1) \tag{13}$$

$$\zeta = \begin{cases} 2\beta & \#E_{i\&c} \geq 1 \\ \beta & \#E_{i\&c} = 0 \end{cases} \tag{14}$$

The specific expression of the function $g(x)$ should not significantly affect the comparison results between the two settings of $\zeta$. For our experiments, we set $g(x) = \frac{1}{e^{\sqrt{x}}}$. We tune both $\alpha$ and $\beta$ within $\{0.25, 0.5, 1, 2, 4\}$. The result is shown in Figure (7).

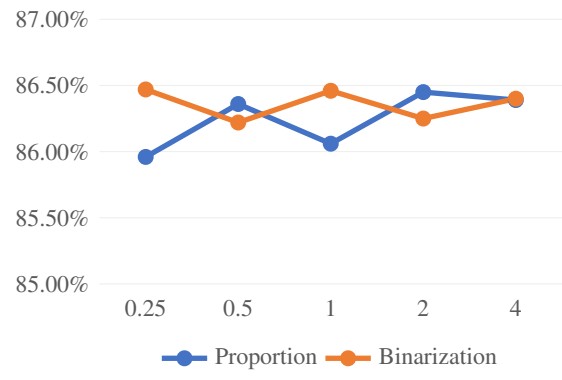

Figure 7: Selection of $\zeta$.

It can be observed that on average, the *Binarization Setting* of $\zeta$ performs slightly better than the *Proportion Setting*, although the difference is not significant. This may be due to the short text in our task, resulting a small number of named entities. Based on these results, we choose the *Binarization Setting* and set $\beta = 1$ since the accuracy is almost the highest while having smaller loss in this case.

## A.2 Selection of $g(x)$

In our final model, we set $g(x) = \frac{1}{e^{\sqrt{x}}}$ since it has the simplest form and a relatively smooth descent. However, we later obtained the performance of different $g(x)$ curves by tuning $a \in \{0, 1, 2, 4, 8\}$ and $b \in \{1, 2\}$, as reported in Figure 8. Interestingly, although not optimal, the $g(x)$ we adopted also exhibits competitive performance. Meanwhile, all $g(x)$ curves outperform the results obtained using binary NEI.

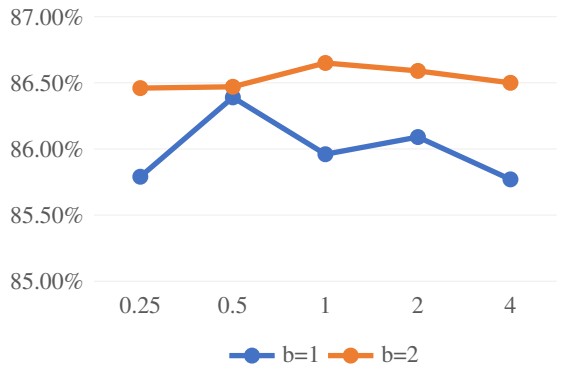

Figure 8: Selection of $g(x)$.

## A.3 Validity Verification

We can discuss the validity of the terms in the SRS by analyzing the impact of removing or modifying them on the performance. When we removed the negative term and only keep the positive term in SRS, the performance drops by 0.22%. Similarly, when we fixed the negative term to 1, the performance drops by 0.24%. These results indicate that the negative term representing refutation relation is important and contribute to the overall performance of our method.

Furthermore, we found that the performance dropped by 0.42% when $g(x)$ was fixed so that it was independent of the input (we set $g(x) = 0.5$). This confirms our observation that named entities with higher frequency of conflicts should express stronger conflicting stance. When the scaling factor $\zeta$ does not scale as $\#E_{i\&c}$ gets smaller (we set $\zeta = 2$), the performance dropped by 0.26%. This indicates that dynamically emphasizing support or refutation relations based on $\#E_{i\&c}$ is effective.

## B Details about Figures 3 and 4

Details about Figures 3 and 4 are presented in Tables 3 and 4, respectively.

|  | Scenario | | | |
|---|---|---|---|---|
|  | a | b | c | d |
| CCN | 81.5% | 90.9% | 78.5% | 84.6% |
| Ours | 85.3% | 92.2% | 81.5% | 89.3% |

Table 3: Performance comparison results between CCN and our method in different OOC scenarios.

|  | Training Set Proportion | | | |
|---|---|---|---|---|
|  | 25% | 50% | 75% | 100% |
| CCN | 77.9% | 80.8% | 82.8% | 83.9% |
| Ours | 83.6% | 85.3% | 86.0% | 87.1% |

Table 4: Performance analysis results in the limited data environment.

## C Multi-Stance Fusion

In addition to the concatenation method, we also tested other methods, including Max-Pooling, Avg-Pooling, Element-wise multiplication, and a combination of the three, followed by a fully connected layer for dimensionality reduction. However, as shown in Table 5, none of these methods were able to achieve the same performance as the concatenation method for stance fusion.

|  | All | Pristine | Falsified |
|---|---|---|---|
| Ours | **87.1%** | 85.5% | **88.6%** |
| Max-Pooling | 86.2% | **86.3%** | 86.0% |
| Avg-Pooling | 85.6% | 84.5% | 86.6% |
| Multiplication | 83.8% | 83.4% | 84.1% |
| All with fc | 86.0% | 84.5% | 87.4% |

Table 5: Performance comparison results of different variants.

## D Ways to Introducing Stance Relation

We notice that some studies try to introduce stance relation into fake news detection. For instance, (a) QSAN (Tian et al., 2020) emphasizes the "opposite" relationship by defining $-$Softmax, and (b) Yao et al. (2022) propsoed to extract stance representations of evidence towards claim in an end-to-end multimodal fact-checking task by two arithmetic operations. For comparison with the way we introduce the stance relation, we integrated the two approaches into the baseline CCN while using the

same features as CCN and our method. Specifically, Equations (15)–(17) demonstrate how we combined approach (a), while Equations (18)–(19) depict how we integrated approach (b).

$$H_*^+ = \text{Softmax}(H^c H_{k*}^e)_{v*}^e + H^c \quad (15)$$

$$H_*^- = -\text{Softmax}(-H^c H_{k*}^e)H_{v*}^e + H^c \quad (16)$$

$$H = W \cdot \text{Concat}(H_*^+, H_*^-) + b \quad (17)$$

$$H_* = \text{Concat}(W_* H_{v*}^e \cdot H^c, W_* H_{v*}^e - H^c) \quad (18)$$

$$H = W \cdot H_* + b \quad (19)$$

| Method | CCN | (a) | (b) | Ours |
|---|---|---|---|---|
| **Accuracy** | 83.9% | 83.9% | 83.8% | 87.1% |

Table 6: Performance comparison results of different ways to introducing stance relation.

The comparison with baseline CCN and our method is shown in Table 6. It can be seen that the accuracy has not significantly improved with the assistance of the two methods, which shows that the introduction of stance in the out-of-context scenario is different from the general fake news scenario.