# OpenReview forum: "Support or Refute: Analyzing the Stance of Evidence to Detect Out-of-Context Mis- and Disinformation"
_EMNLP/2023/Conference — EMNLP 2023 Main_

### Official Review · Reviewer_RUdv · 2023-08-04

**Soundness:** 4

**Excitement:**

4: Strong: This paper deepens the understanding of some phenomenon or lowers the barriers to an existing research direction.

**Paper Topic And Main Contributions:**

The study takes an important social issue, "detecting out of context misuse of images to spread online misinformation," and proposes a multimodal processing framework of multimodal evidence and content. The paper accounts for text and visual evidence and their relationship with the image-caption input pairs in detecting out of context misinformation in the input by proposing a stance extraction network. The authors provide comparison with several baselines on the NewsCLIPpings dataset and present ablation studies and analyses that support the effectiveness of their framework in addressing out of context multimodal misinformation. This paper also includes code for reproducing its findings.

**Questions For The Authors:**

You have provided a good set of comparisons with other baselines, however you have not included COSMOS by Aneja et. al. (2021) (https://github.com/shivangi-aneja/COSMOS) which you cite as well. Given how close the two papers are (with the exception of COSMOS not taking into account external evidence), was there a reason a comparison was not made?

**Reasons To Accept:**

- Paper addresses an important social problem: finding repurposed images that may spread online misinformation.

- Sound methodology and study design; thorough analyses of results and model capabilities.

- Submission includes code for reproduciblity

- Reasonable and accurate description of limitations and ethical considerations around the proposed method.

**Reasons To Reject:**

I have included some suggestions for improving the manuscript, but none are major enough to be a cause for rejection IMO.

**Reproducibility:**

4: Could mostly reproduce the results, but there may be some variation because of sample variance or minor variations in their interpretation of the protocol or method.

**Reviewer Confidence:**

4: Quite sure. I tried to check the important points carefully. It's unlikely, though conceivable, that I missed something that should affect my ratings.

**Typos Grammar Style And Presentation Improvements:**

What is the difference between misinformation and disinformation, in the context of your paper? Does your model processes misinformation and disinformation differently? if not, it may be helpful to just use one term to describe them, as is commonly done in the literature when the two are not differentiated by the method.

SOTA (state of the art) is spelled incorrectly as STOA (lines 428 and 429)

---

> ### Author Rebuttal · Authors · 2023-08-28
>
> We thank the reviewer for their positive and insightful feedback. Below please find our responses to issues and concerns raised by the reviewer.
>
> **Questions For The Authors: You have provided a good set of comparisons with other baselines, however you have not included COSMOS by Aneja et. al. (2021) (https://github.com/shivangi-aneja/COSMOS) which you cite as well. Given how close the two papers are (with the exception of COSMOS not taking into account external evidence), was there a reason a comparison was not made?**
>
> **Our response:** COSMOS is indeed related to OOC image-caption mismatch detection. However, the input of COSMOS is conceptually different from our OOC image-caption pair detection: the input of COSMOS a triplet (image, caption1, caption2) and it tries to detect if the two captions associated with the same image are semantically incoherent, which indicates that either caption1 or caption2 is the consequence of an OOC image-caption pair. However, our method takes one image and one caption as the input and predict if this given image-caption pair is pristine or OOC. These two detection tasks are not directly comparable, although our method can be generalized to the task COSMOS is designed for. Note that other researchers working on OOC also mentioned such difference in their papers, e.g., the NewsCLIPPings paper and the CCN paper. We do have a plan to generalize our method and compare how it can be used to detect incoherence in a triplet (image, caption1, caption2) like what COSMOS does, but we leave this as our future work and will add it to our final paper.
>
> **Typos Grammar Style And Presentation Improvements: What is the difference between misinformation and disinformation, in the context of your paper? Does your model processes misinformation and disinformation differently? if not, it may be helpful to just use one term to describe them, as is commonly done in the literature when the two are not differentiated by the method.**
>
> **Our response:** Our method can work with both mis- and disinformation as long as the image and the caption does not match. While OOC image-caption pairs are more likely disinformation in the first instance (i.e., the creator has a clear intention to cheat, although in rare cases the creator may accidentally mismatched an image with a wrong caption), some people may spread an OOC image-caption pair without knowing that it is false -- in this case the false OOC information will be more like misinformation (i.e., the spreader has no intention to cheat). We will clarify this point in our final paper and consider using the more neutral term "OOC false information" to replace the potentially confusing term "OOC mis-/disinformation".
>
> **Typos Grammar Style And Presentation Improvements: SOTA (state of the art) is spelled incorrectly as STOA (lines 428 and 429)**
>
> **A3:** Thanks for pointing out this typo. We will carefully proofread the paper again to identify and fix as many such mistakes as possible for the final edition.

---

### Official Review · Reviewer_V426 · 2023-08-04

**Typos Grammar Style And Presentation Improvements:** Figure 1 Textual Evidence is misspell…
**Soundness:** 4

**Excitement:**

4: Strong: This paper deepens the understanding of some phenomenon or lowers the barriers to an existing research direction.

**Paper Topic And Main Contributions:**

This study thoroughly investigates the widespread problem of mis- and disinformation on the internet, focusing particularly on out-of-context (OOC) information. This occurs when unrelated pieces of information are falsely linked, resulting in a misleading combination of genuine images and deceitful textual descriptions.
In response to this issue, the researchers introduce an innovative solution called the Stance Extraction Network (SEN). The SEN is a comprehensive framework, purpose-built to distill the stances from a wide range of multi-modal evidence. This approach signifies an evolution in the study of OOC misinformation by appreciating the multi-faceted nature of the evidence involved and recognizing their inherent biases and unique perspectives.
Furthermore, the research presents a novel integration of a support-refutation score into the textual SEN, which is a metric derived from the co-occurrence relations of named entities. This refinement amplifies the depth of analysis possible for textual misinformation, enhancing the overall robustness of the method and providing interpretability.
The researchers subject their proposed solution to rigorous empirical testing using an expansive public dataset. The resulting data underscores the superiority of the new method compared to existing state-of-the-art techniques, with the highest-performing model recording a performance increment of 3.2%.

**Questions For The Authors:**

A. Why did your data sample your training data? Did you re-train and re-evaluate other models in your sample environment? Would this shift in the dataset change the model's performance, either in data efficacy or overall performance?

**Reasons To Accept:**

1. This paper proposes an innovative approach to one of the most prominent problems in the digital world and is highly relevant to society.
2. Their method not only detects mis/disinformation effectively but also provides interpretability, which is helpful for experts or the audience.
3. This method surpasses the existing best approach in a pressing issue.
4. They provide extensive analysis of their model and comparison to show their model performance.

**Reasons To Reject:**

1. The paper writing is too dense and somewhat hard to follow.
2. In the paragraph starting at line 470, the paper has compared their method to the CLIP approach, which performs considerably lower than theirs. (In Figures 3 & 4, the model's name shows CNN, but in the text, it refers to the CLIP model). Comparing must be done your model with the existing best model on OOC scenarios.
3. The scenarios (a, b, c, d) are unclear. It makes the analysis lines 476 and 482 hard to follow.
4. The conclusion of the two-sample one-tail z-test is not clearly explained.
5. The data sampling is indefinite. The chosen samples can affect the evaluation outcome.

**Reproducibility:**

4: Could mostly reproduce the results, but there may be some variation because of sample variance or minor variations in their interpretation of the protocol or method.

**Reviewer Confidence:**

2: Willing to defend my evaluation, but it is fairly likely that I missed some details, didn't understand some central points, or can't be sure about the novelty of the work.

---

> ### Author Rebuttal · Authors · 2023-08-28
>
> We thank the reviewer for their thoughtful feedback. Below please find our responses to issues and concerns raised by the reviewer.
>
> **Reasons To Reject 1: The paper writing is too dense and somewhat hard to follow.**
>
> **Our response:** We thank the reviewer's comment, and we will conduct a more careful proofreading to see if the writing can be further improved.
>
> **Reasons To Reject 2: In the paragraph starting at line 470, the paper has compared their method to the CLIP approach, which performs considerably lower than theirs. (In Figures 3&4, the model's name shows CCN, but in the text, it refers to the CLIP model). Comparing must be done your model with the existing best model on OOC scenarios.**
>
> **Our response:** Thanks for spotting this inconsistency. We made a mistake in the main text. The model we compared our method with is indeed the CCN approach, the best existing model. We will fix this error in the final paper.
>
> **Reasons To Reject 3: The scenarios (a, b, c, d) are unclear. It makes the analysis lines 476 and 482 hard to follow.**
>
> **Our response:** Thanks for pointing out that the descriptions about the four scenarios are not sufficiently clear. Note that our proposed method outperforms the best baseline CCN in all the four scenarios, although we decided to focuses more on the most challenging Scenario c. In the following, we clarify more about the four scenarios and why we focused on Scenario c.
>
> The four different OOC scenarios were originally defined by authors of the NewsCLIPpings paper. They defined the scenarios in a relatively complicated way, so that they had to use more than two pages for the descriptions. For our paper, due to the limited space, we decided to focus on the most challenging Scenario c in lines 481-484 and explain other three scenarios only briefly. If the paper is accepted, we will add the following more detailed descriptions of all the four scenarios to the final paper (which we understand is possible because accepted papers will be allowed to have one more page for the main body).
>
> * Both Scenarios a and b, i.e., Semantics/CLIP Text-Image and Semantics/CLIP Text-Text, are about attempts of portraying the subject of images as other named entities and use CLIP embeddings for mismatching. The main difference is that Scenario a tries to form OOC pairs by finding non-matching image-caption pairs with the highest similarity directly, while Scenario b by first finding non-matching caption-caption pairs with the highest similarity and then using the corresponding image of the second caption.
>
> * Scenario c, i.e., Person/SBERT-WK Text-Text, portrays the same person in a false context, such as the wrong events or places. It uses the lowest SBERT-WK text-text similarity to ensure the person appearing in different contexts.
>
> * Scenario d, i.e., Scene/ResNet Place, describes the event in the image as another event of the same event type.
>
> With these explanations, we believe that the analysis between lines 476 and 482 will be much easier to understand. Since Scenario b uses the similarity of caption-caption instead of the similarity of image-caption for mismatching, it is relatively easy to detect in a multi-modal framework. Due to the special mismatching method of Scenario c, only models with a strong semantic understanding ability can discover the subtle differences in the context, so we consider it more difficult to detect.
>
> **Reasons To Reject 4: The conclusion of the two-sample one-tail z-test is not clearly explained.**
>
> **Our response:** We have shown in lines 538-540 that there are significant differences in the distribution of pristine and OOC data on SRS, and explained in lines 544-546 that the purpose of conducting the two-sample one-tail z-test is to statistically verify this observable distribution difference. After reading the description again, we now feel some parts could benefit from more explanation, e.g., how the value of gamma was handled and how the statistical significance changes w.r.t. the value of gamma. We will add such additional discussions to the final paper.
>
> **Reasons To Reject 5: The data sampling is indefinite. The chosen samples can affect the evaluation outcome.**
>
> **Our response:** We understand that the reviewer refers to the results based on a sampled subset of the whole training dataset, reported in Section 4.4. We used three sampling rates (25%, 50% and 75%) and our purpose is to study how well our method works when the training set available is smaller. This is useful because if a model can work with a smaller amount of training data, then the efforts needed to gather and label training data will be lower. As shown in Figure 4, compared with the CCN method, our proposed method can have an even higher performance gain with a smaller number of training samples, indicating that our model can learn more effectively and more efficiently (i.e., faster) than the CCN model. In the final paper, we can include a new table to show the precise performance gains under the four different sampling rates (25%, 50%, 75% and 100%). For the reproducibility purposes, we saved the random sampling data we used, although we did not include the data in the supplementary materials. In order to facilitate reprodicibility of our results, we will release all such data together with our source code once our paper is accepted. Finally, we would like to clarify that in Section 4.3, all our results are based on the whole training dataset, which give the best results for all methods, but our proposed method remains the best.
>
> **Question For The Authors A: Why did your data sample your training data? Did you re-train and re-evaluate other models in your sample environment? Would this shift in the dataset change the model's performance, either in data efficacy or overall performance?**
>
> **Our response:** Regarding why we sampled the training dataset, we have explained above. We re-trained and re-evaluated all the models in every sampling environment. The performance of the model in a limited data environment can be seen in Figure 4. While the reduced amount of training data does reduce the accuracy of all models, but our model demonstrated an even higher performance gain relatively to the best existing model (CCN).
>
> **Typos Grammar Style And Presentation Improvements: Figure 1 Textual Evidence is misspelled. Line 136, metadata is misspelled.**
>
> **Our response:** Thanks for spotting such typos. We will fix them.
>
> **Soundness: 3: Good: This study provides sufficient support for its major claims/arguments, some minor points may need extra support or details.**
>
> **Our response:** We hope the above responses have clarified those minor points.
>
> **Reproducibility: 4: Could mostly reproduce the results, but there may be some variation because of sample variance or minor variations in their interpretation of the protocol or method.**
>
> **Our response:** Hope the above response on the data sampling has clarified this reproducibility point.

---

### Official Review · Reviewer_2aq7 · 2023-08-07

**Soundness:** 4

**Excitement:**

4: Strong: This paper deepens the understanding of some phenomenon or lowers the barriers to an existing research direction.

**Paper Topic And Main Contributions:**

The paper presents a framework for automatic out-of-context mis-/disinformation detection using the stance (support/refutes) of multi-modal evidences (texts and images). The authors' proposed architecture leverages stance extraction networks for semantic stance comparisons between image and visual evidence. Finally, they incorporate support-refutation score calculated based on the co-occurrence relationship of named entities. The proposed approach outperforms strong state-of-the-art baselines on the NewsCLIPpings datasets.

**Reasons To Accept:**

1. Multimodal mis-/disinformation detection is an important and timely topic.
2. The proposed approach models simultaneously the textual and visual evidences using multi-stance fusion.
3. The proposed framework outperforms significantly state-of-the-art approaches both in low-resource and full-resource setting.

**Reasons To Reject:**

1. The proposed method is fairly complex and depends on multiple networks and an external NER component. However, the authors show that all of the components are important for the results
2. I'm not fully convinced by the explainablity analysis presented by the authors (also the explanbility of the model itself). More precisely, even though the two distributions differ in terms of support-refute score this is a feature in the model and it unclear how it correlates with the predicted label (pristine/out-of-context).

**Reproducibility:**

3: Could reproduce the results with some difficulty. The settings of parameters are underspecified or subjectively determined; the training/evaluation data are not widely available.

**Reviewer Confidence:**

3: Pretty sure, but there's a chance I missed something. Although I have a good feel for this area in general, I did not carefully check the paper's details, e.g., the math, experimental design, or novelty.

**Typos Grammar Style And Presentation Improvements:**

* L428 STOA -> SOTA
* The metric is unclear in: L026 "a performance gain of 3.2%." and in the caption of Table 1.

---

> ### Author Rebuttal · Authors · 2023-08-28
>
> We thank the reviewer for their constructive feedback. Below please find our responses to issues and concerns raised by the reviewer.
>
> **Reason To Reject 1: The proposed method is fairly complex and depends on multiple networks and an external NER component. However, the authors show that all of the components are important for the results.**
>
> **Our response:** Although our proposed method is indeed quite complex, we would like to highlight that it uses the same number of networks as some of the SOTA baselines such as the CCN method. As the reviewer pointed out, we include all the components to help improve the performance of the designed classifier beyond other SOTA methods, therefore the high complexity is just a consequence of including such components.
>
> **Reason To Reject 2: I'm not fully convinced by the explainablity analysis presented by the authors (also the explainbility of the model itself). More precisely, even though the two distributions differ in terms of support-refute score this is a feature in the model and it unclear how it correlates with the predicted label (pristine/out-of-context).**
>
> **Our response:** Thanks for pointing out the lack of clarity of the explainability subsection. The effect of the SRS feature can be seen from the ablation analysis shown in Table 2: when we remove the SRS feature (w/o SRS) or further simplify it into a binary feature (binary NEI), the accuracy will drop by 2.5% and 1.4%, respectively. We will refine the subsection by adding this analysis in.
>
> **Typos Grammar Style And Presentation Improvements**
>
> **Our response:** We thank the reviewer for the two issues and will fix them and do further proofreading to correct any other typos, grammatical errors and presentation issues.
>
> **Reproducibility: 3: Could reproduce the results with some difficulty. The settings of parameters are underspecified or subjectively determined; the training/evaluation data are not widely available.**
>
> **Our response:** We appreciate the reviewer's concerns on the difficulty about reproducibility. We have released all the code in the supplementary materials, and README.md contains the steps and commands required to replicate our experimental results. The datasets we used are all public ones so readers can easily access them and run our code against the datasets. If the paper is accepted, we will release the code and relevant documents on GitHub as a public resource to facilitate reproduction of our experimental results. We will also release the random samples used in our experiments to allow reproducing the results reported in this paper. We also welcome any further advice from the reviewer on what we can do to further improve reproducbility of our work.

---

### Meta-Review · Area_Chair_kbFs · 2023-09-19

**Recommendation:** 5

**Metareview:**

This research focuses on out-of-context (OOC) mis-/disinformation on the Web and proposes a method to predict it using evidence and stances consisting of text and images.

Pros:
Mis-/disinformation identification is an urgent issue, and the effort is commendable.
The approach is based on a good understanding of the nature of the problem.

Cons:
Key questions from reviewers have been addressed in the rebuttal, but these need to be properly reflected in the camera-ready version.

---

### Decision · Program_Chairs · 2023-10-07

**Decision:**

Accept-Main

**Comment:**

This research focuses on out-of-context (OOC) mis-/disinformation on the Web and proposes a method to predict it using evidence and stances consisting of text and images.

Pros:
Mis-/disinformation identification is an urgent issue, and the effort is commendable.
The approach is based on a good understanding of the nature of the problem.

Cons:
Key questions from reviewers have been addressed in the rebuttal, but these need to be properly reflected in the camera-ready version.